# Aerobic Degradation Characteristics of Decabromodiphenyl ether through *Rhodococcus ruber* TAW-CT127 and Its Preliminary Genome Analysis

**DOI:** 10.3390/microorganisms10071441

**Published:** 2022-07-17

**Authors:** Hao Xu, Qingtao Cai, Qiuying An, Chen Tang, Wanpeng Wang, Guangshun Wang, Wanting You, Dongbei Guo, Ran Zhao

**Affiliations:** 1State Key Laboratory of Molecular Vaccinology and Molecular Diagnostics, School of Public Health, Xiamen University, Xiamen 361102, China; xuhaosmail@163.com (H.X.); aqy211827@163.com (Q.A.); tangchen@stu.xmu.edu.cn (C.T.); wgs13860163044@163.com (G.W.); ywttoo@163.com (W.Y.); guodb@xmu.edu.cn (D.G.); 2The Second People’s Hospital of Futian District in Shenzhen, Shenzhen 518000, China; caiqingtao1@126.com; 3State Key Laboratory Breeding Base of Marine Genetic Resources, Third Institute of Oceanography, Ministry of Natural Resources, Xiamen 361005, China; wangwanpeng@tio.org.cn

**Keywords:** BDE-209, *Rhodococcus ruber*, TAW-CT127, aerobic degradation, genomics

## Abstract

Decabromodiphenyl ether (BDE-209), a polybrominated diphenyl ether (PBDE) homolog, seriously threatens human health. In this study, a *Rhodococcus ruber* strain with high BDE-209 degradation activity, named TAW-CT127, was isolated from Tong’an Bay, Xiamen. Under laboratory conditions, the strain’s optimal growth temperature, pH, and salinity are 45 °C, 7.0, and 0–2.5%, respectively. Scanning electron microscopy (SEM) analysis shows that TAW-CT127 is damaged when grown in manual marine culture (MMC) medium with BDE-209 as the sole carbon source instead of eutrophic conditions. In the dark, under the conditions of 28 °C, 160 rpm, and 3 g/L (wet weight) TAW-CT127, the degradation rate of 50 mg/L BDE-209 is 81.07%. The intermediate metabolites are hexabromo-, octabromo-, and nonabromo-diphenyl ethers. Through whole-genome sequencing, multiple dehalogenases were found in the genome of TAW-CT127; these may be involved in the production of lower-brominated diphenyl ethers. Additionally, biphenyl-2,3-dioxygenase (BDO) in TAW-CT127 may catalyze the debromination reaction of BDE-209. Our research provides a new high-efficiency strain for bioremediation of BDE-209 pollution, and lays the foundation for the preliminary exploration of genes associated with BDE-209 degradation.

## 1. Introduction

Polybrominated diphenyl ethers (PBDEs) are listed as typical persistent organic pollutants according to the Stockholm Convention in 2009, and are extensively used in a range of fields (electronic appliances, textiles, construction, etc.) to reduce risk of fire and improve flame retardancy [1]. These pollutants are easily released into the environment to varying degrees during production, transport, addition to products, and waste storage and disposal, thereby posing threats to ecosystems. Owing to their environmental persistence, lipophilicity, and bioaccumulation, PBDEs have been detected in various environmental media, such as water [2,3], dust [4], soil [5], the atmosphere [6,7,8], birds [9], aquatic organisms [10], and human bodies [11]. Toxicological experiments show that PBDEs disrupt the reproductive and immune systems of humans and their offspring, mainly by altering the hormonal system and causing negative embryonic developmental and carcinogenic effects [12]. Global environmental pollution caused by the extensive use of PBDEs has attracted widespread attention.

Although PBDEs consist of 209 congeners, there are three major commercial mixtures: pentabromodiphenyl ether (penta-BDE), octabromodiphenyl ether (octa-BDE), and decabromodiphenyl ether (BDE-209) [1,13]. Penta- and octa-BDE are more toxic than other PBDE homologues, causing significant biological toxicity; therefore, commercial penta- and octa-BDEs were banned in North America in 2004 [12,14,15]. However, BDE-209 is not subjected to regulatory control in China [16]. BDE-209 is the only commercial PBDE that has not been globally banned, owing to its high efficiency of flame retardancy and low biotoxicity. However, it was reported that BDE-209 leads to reproductive toxicity by inducing telomere dysfunction and related cell senescence and apoptosis in the testes of Sprague–Dawley rats [17]. BDE-209 causes gut toxicity by modulating several properties of the intestinal barrier, including permeability, oxidative stress, autophagy, apoptosis, and inflammation [18]. BDE-209 can enter the body via respiratory, oral, and dermal routes [19]. People who work for long periods of time producing BDE-209 displayed serum and urine levels of BDE-209 that show a significant positive correlation with thyroid hormone levels; this in turn can lead to symptoms of hyperthyroidism [20]. Li et al. found that exposure to BDE-209 results in genome-wide methylation in mouse GC-2spd cells, further affecting cell growth and possibly causing reproductive toxicity [21]. In addition, BDE-209 toxicity includes hepatotoxicity [22], nephrotoxicity [23], embryonic developmental toxicity [24], carcinogenicity [25], and potential neurodevelopmental toxicity in humans, especially in children [4]. Studies show that BDE-209 not only induces the above-mentioned toxicities, but may also produce other lower brominated diphenyl ether congeners, such as penta-BDE and octa-BDE [26]. Therefore, environmental pollution by BDE-209 has attracted global attention in recent years. Environmentally friendly techniques that can effectively remove BDE-209 from environmental media play an important role in solving this pollution problem.

The main treatment techniques for BDE-209 include photodegradation [27,28], chemical degradation [29,30,31], and microbial degradation [32]. Microbial degradation remediation technology uses the degradative abilities of microorganisms to reduce or eliminate BDE-209 from the environment. Compared to traditional physical and chemical remediation methods, microbial processes have many advantages, including high efficiency, short time consumption, low cost, and environmental friendliness [33]. In recent years, some strains with the ability to degrade BDE-209 were identified, such as *Stenotrophomonas sp.* [34], *Rhodococcus sp.* [35], *Bacillus tequilensis* [36], and *E. casseliflavus* [37]. However, most bacteria exhibit low degradation efficiency and environmental adaptability. One study reports that *B. Brevis* exhibits a low degradation efficiency of up to 55% in 5 d, under 0.5 mg/L BDE-209 by 1 g/L strain inoculum [38]. Another study reports that *P. aeruginosa* exhibits superior degradation efficiency, but a high concentration of BDE-209 limits growth of the strain (5–10 mg/L) [39]. Since few strains with high salt tolerance and high degradation efficiency are reported, the study of aerobic microbial degradation of BDE-209 has important theoretical significance for the remediation of this ecological pollutant.

In this study, *Rhodococcus ruber* strain TAW-CT127 was isolated, purified, and characterized for the aerobic degradation of BDE-209, with high adaptability and degradation efficiency. The effects of different growth conditions including temperature, pH, and salinity were determined. The degradation effects, products, and whole-genome sequence of the strain were also studied to explore the degradation mechanism. The present findings enrich the study of microbial remediation flora for BDE-209, and provide possible strategies to ameliorate the environmental pollution of hazardous PBDEs.

## 2. Materials and Methods

### 2.1. Chemicals and Media

BDE-209 (98% purity) was purchased from Sigma-Aldrich (St. Louis, MO, USA). Standards for octa-BDE, penta-BDE, and BDE-209 were purchased from the National Academy of Metrology (Beijing, China). The following solvents were used for the extraction, clean-up, and analysis of metabolites: n-hexane, ethyl acetate, dichloromethane, and acetone. The manual marine culture (MMC) medium used for bacterial growth and isolation contained NaCl (24.0 g/L), MgSO_4_·7H_2_O (7.0 g/L), NH_4_NO_3_ (1.0 g/L), KCl (0.7 g/L), para (2.0 g/L), and Na_2_HPO_4_ (3.0 g/L). This was adjusted to pH 7.4 ± 0.2, and an appropriate amount of trace element mixture was added after sterilization. The trace element mixture contained CaCl_2_ (2.0 mg/L), FeCl_3_·6H_2_O (50.0 mg/L), CuSO₄ (0.5 mg/L), MgCl_2_·4H_2_O (0.5 mg/L), and ZnSO_4_·7H_2_O (10.0 mg/L), and was sterilized by passing it through a 0.22 μm filter membrane. The marine 2216 Luria–Bertani (marine 2216 LB) medium used to enrich bacteria contained NaCl (19.45 g/L), MgCl_2_ (8.8 g/L), peptone (5.0 g/L), Na_2_SO_4_ (3.24 g/L), CaCl_2_ (1.8 g/L), yeast extract (1.0 g/L), KCl (0.55 g/L), NaHCO_3_ (0.16 g/L), ferric citrate (0.1 g/L), KBr (0.08 mg/L), SrCl_2_ (34.0 mg/L), H_3_BO_3_ (22.0 mg/L), Na_2_HPO_4_ (8.0 mg/L), Na_2_SiO_3_·7H_2_O (4.0 mg/L), NaF (2.4 mg/L), and NH_4_NO_3_ (1.6 mg/L). The pH of this medium was adjusted to 7.6 ± 0.2. Marine 2216 LB agar plates included 1.5% (*w*/*v*) agar. All culture media were sterilized in an autoclave (121 °C, 20 min) before use. The above and other reagents were obtained from the Sinopharm Group Chemical Reagent Co. (Beijing, China).

### 2.2. Isolation of Strain TAW-CT127

Sediment samples were collected from a depth of 2 m in Tong’an Bay, Xiamen, Fujian Province, China (24° N, 118° E). For enrichment, 100 mL of MMC medium was inoculated with approximately 5 g of sediment sample and 100 mg/L of BDE-209 as the sole carbon source. This was then incubated in a shaker at 28 °C and 160 rpm in the dark for four cycles, each lasting four weeks. After 16 weeks, the enriched suspension was serially diluted to 10^−10^ of the initial concentration and inoculated on marine 2216 LB agar medium. The plates were incubated at 28 °C for three days. After 3 days of incubation, individual colonies were picked and purified by successive streaking. The isolated bacterium was designated as a strain with high BDE-209 degradation efficiency for further study.

### 2.3. Identification of Strain TAW-CT127

Molecular identification was performed by extracting genomic DNA using a DNA isolation kit (SBS Genetech Co., Ltd., Beijing, China). Furthermore, the isolated 16S rRNA template was amplified by polymerase chain reaction (PCR) using the primers 27F (5′-AGAGTTTGATCMTGGCTCAG-3′) and 1492R (5′-TACGGYTACCTTGTTACGACTT-3′) [40]. The amplicons were purified using a PCR product purification and recovery kit from AxyPrep™ (Shanghai, China) and sent to Majorbio (Shanghai, China) for sequencing. The results were submitted to the National Center for Biotechnology Information (NCBI) gene database (https://submit.ncbi.nlm.nih.gov/subs/genbank/, (Submission number: SUB11805222)), and 16S rRNA sequences were obtained from the NCBI nucleotide database using the basic local alignment search tool (BLAST) function. The gene sequence of strain TAW-CT127 is listed in Appendix A. Finally, the phylogenetic tree of this strain was constructed with MEGA5.0 using the neighbor-joining method.

### 2.4. Morphological Observation and Physiological and Biochemical Characteristics of the Strain

Under aseptic conditions, bacterial TAW-CT127 strain cells growing on marine 2216 LB agar plates were observed under a light microscope (IX70, Olympus, Tokyo, Japan) to evaluate the shape, size, color, transparency, and edge characteristics of the colonies.

Physiological and biochemical analyses were performed according to the environmental engineering microorganism detection manual [41,42]. Gram staining experiments were performed, based on the method described by Li et al. [43]. Physiological and biochemical characterizations including the Voges–Proskauer (VP), starch hydrolysis, glucose oxidative fermentation, gelatin hydrolysis, peroxidase, and contact oxidase tests were performed. Strain enzyme activity tests were performed using the API ZYM enzyme activity kit (BioMerieux, Marcy l’ Etoile, France). In addition, strain biochemical characterization experiments were performed using the API 20E bacterial biochemical test kit (BioMerieux, Marcy l’Etoile, France). Other than that, experiments to determine the utilization of different substrates by the strain were performed using the API 20NE bacterial metabolism test kit (BioMerieux).

### 2.5. Optimal Growth Conditions for Strain TAW-CT127

The TAW-CT127 strain was cultured in a 150 mL Erlenmeyer flask containing 100 mL marine 2216 LB medium. The cells were then harvested by centrifugation at 6000 rpm for 10 min, and washed three times with 20 mM sterile phosphate-buffered saline (PBS) solution. The bacteria were collected after the third wash and resuspended in 20 mM sterile PBS solution, so that the OD600 of the resuspension was between 0.4–0.5. The resuspension was then inoculated with marine 2216 LB medium at a 3 g/L (wet weight) inoculum for subsequent determination of optimal growth conditions for strain TAW-CT127. The bacterial growth conditions were evaluated at different temperatures, pH levels, and salinity. The different temperatures tested were as follows: 4, 15, 20, 25, 30, 35, 40, and 45 °C. The effects of different pH levels from 3.0 to 12.0 (in increments of 1 pH unit) were tested, and adjusted with 1 M HCl or 1 M NaOH in marine 2216 LB medium. pH adjustment was performed after sterilization of the medium. For sterilization, 1 M HCl or 1 M NaOH was filtered through a 0.22 μm membrane. The different salinities (0, 0.5, 1.0, 1.5, 2.0, 2.5, 3.0, 3.5, 4, 6, 8, 12, 16, 20, and 25%; *m*/*v*) tested were prepared according to the formula, but without NaCl.

Measurement of the optical density of the bacterial solution at OD600 was performed using a full wavelength enzyme labeler (Epoch 2, BioTek, Winooski, VT, USA). Bacterial samples were taken at 0 and 48 h of incubation, and growth rate and optical density were measured at different temperatures, pH, and salinities. All samples were tested in triplicate.

### 2.6. Observation of the Surface Morphology of TAW-CT127 under Different Culture Conditions

The bacterial resuspension was collected as described in Section 2.5. The prepared suspensions were inoculated with marine 2216 LB and MMC media, with BDE-209 (50 mg/L) as the sole carbon source at 1% (*v/v*) and 3 g/L (wet weight). The medium was placed on a constant-temperature shaker (28 °C, 160 rpm) for 5 days in the dark. The thallus were obtained after centrifugation, and washed twice with 100 mM sterile PBS solution at 2000 rpm for 30 min each. A total of 30 μL of the prepared bacterial solution was taken, dried, sprinkled with gold, and finally observed using scanning electron microscopy (SEM) analysis (Apreo S, Thermo Fisher Scientific, Waltham, MA, USA).

### 2.7. Degradation Characteristics of BDE-209 by Strain TAW-CT127

#### 2.7.1. Degradation Rate of BDE-209 by TAW-CT127

Ten milligrams of BDE-209 was weighed into a 100 mL volumetric flask, and chromatographic grade dichloromethane was added. Ultrasound was used to assist with dissolution for 30 min; ice was added continuously during this process to prevent loss of BDE-209 due to increased temperature. A BDE-209 stock solution with a concentration of 100 mg/L was prepared, and an appropriate amount was added to a 50 mL sterile centrifuge tube and left until the dichloromethane had completely evaporated. The control group was supplemented with MMC medium only, and the experimental group was supplemented with MMC medium and TAW-CT127 strain resuspension, so that the initial concentration of BDE-209 was 50 mg/L.

Aerobic degradation of BDE-209 was determined at 28 °C in a MMC medium containing either 50 mg/L BDE-209 and 3.0 g/L (wet weight) TAW-CT127, or BDE-209 only, by shaking at 160 rpm for 5 days in the dark. The medium without the bacterium was used as a control, and all experiments were performed in triplicate. To extract BDE-209, the bacterial liquid sample was adjusted to pH 2.0, and then the sample was removed twice using an equal volume of dichloromethane and n-hexane (1:1, *v/v*), assisted by sonication. Anhydrous sodium sulfate (650 °C, 4 h) was added for dehydration. Subsequently, organic extracts were combined and samples were concentrated using a rotary evaporation apparatus (Re-5003, Yuhui, Xian, China) in a water bath at 40 °C. Then, the flask was washed several times with chromatogram class dichloromethane and fixed at 10 mL, ready for the analysis.

For the quantification and metabolite analysis of BDE-209, gas chromatography–mass spectrometry (GC–MS; 6890N/5975, Agilent, Santa Clara, CA, USA) with an electron ionization (EI) source was used. A DB-5MS column (15 m × 0.250 mm × 0.250 μm, Agilent, USA) was used with helium (≥99.999% purity) as carrier gas, and the flow rate was set at 1 mL/min. The injection volume was 1 μL, and the injection method was pulsed non-split injection. The column temperature was initially set at 120 °C and held for 2 min, after which it was ramped up to 250 °C at 40 °C/min, and then to 310 °C at 10 °C/min. The ion source temperature was set to 230 °C, the quadrupole temperature was 150 °C, and the GC–MS interface temperature was held at 280°C. The quantitative analysis was performed in SIM mode with 799.6 quantitative ions. The degradation rate of BDE-209 in the control and experimental groups was calculated by the following formula:(1)$$\mathrm{Degradation}\;\mathrm{rate}\;(\%)=\;\frac{{\mathrm{Initial}\;\mathrm{amount}}_{\mathrm{BDE}\text{-}209}-{\mathrm{Residual}\;\mathrm{amount}}_{\mathrm{BDE}\text{-}209}\;}{{\mathrm{Initial}\;\mathrm{amount}}_{\mathrm{BDE}\text{-}209}\;}\times 100$$

#### 2.7.2. Detection of BDE-209 Metabolites

The samples were collected as described in Section 2.7.1. The samples were placed on ice, the cells were broken at low temperature for 10 min, and the bacterial liquid sample was adjusted to pH 2.0. Liquid–liquid extraction was performed four times; the first two extractants were ethyl acetate (4°C pre-cooling), and the last two extractants were dichloromethane/n-hexane mixed (1:1, *v*/*v*). Organic extracts were combined and samples were concentrated using a rotary evaporation apparatus (Re-5003, Yuhui, Xian, China) in a water bath at 40 °C. The organic components were collected using chromatographically pure dichloromethane, and concentrated to 400 μL using the nitrogen blowing method. BSTFA silylation reagent was added, and the solution incubated in a water bath at 60 °C for 1 h before being analyzed for metabolites using GC–MS. The qualitative analysis was performed in SCAN mode (*m*/*z* 100–1000) with 231.8, 399.7, 799.6, and 957.5 qualitative ions. The column temperature for qualitative analysis was initially set at 80 °C and held for 3 min, after which it was ramped up to 310 °C at a rate of 10 °C/min. Otherwise, GC–MS analysis was performed as in Section 2.7.1.

### 2.8. Genomic Sequencing and Analysis of TAW-CT127

#### 2.8.1. DNA Extraction, Library Construction, and Genome Sequencing

Genomic DNA was extracted from TAW-CT127 using an SBS Genetech DNA isolation kit (Beijing, China) according to the manufacturer’s instructions. Whole-genome sequence analysis of TAW-CT127 was performed using the Illumina Miseq sequencing platform from MajorBio (Shanghai, China) to construct an Illumina paired-ends (PE) library (5000 bp). The whole-genome scan of the strain was completed after quality control of the obtained sequencing data.

#### 2.8.2. Genome Assembly and Functional Annotation

Glimmer software (version 3.02) was used for coding sequence prediction. The predicted coding sequences were annotated from the Clusters of Orthologous Groups (COG) and Kyoto Encyclopedia of Genes and Genomes (KEGG) databases, using BLAST. Each set of query proteins was aligned with the databases, and the best-matched subjects (*e*-value < 10^−5^) were obtained for gene annotation.

### 2.9. Statistical Analysis

Statistical significance was evaluated using SPSS (version 23.0, IBM, Armonk, NY, USA). One-way ANOVA and the Tamhan test were applied to test for significance at *p* < 0.05.

## 3. Results and Discussion

### 3.1. Morphological Observation, Physiological and Biochemical Characteristics, and Optimum Growth Conditions of the Strain

After four enrichment cycles, isolation, and purification on Marine 2216 medium, a single strain was labeled TAW-CT127. TAW-CT127 is found to be a Gram-positive bacterium, and its colonies are orange, circular, wet, smooth, and opaque, with raised center and regular edges when grown on marine 2216 agar incubated at 28 °C for 3 days (Figure 1). In addition, TAW-CT127 produces catalase, but not starch hydrolase or contact oxidase, and the VP test is negative. Using the API 20E system, it is determined that the strain displays the ability to utilize citrate, and produce acetyl methyl methanol without the production of H_2_S. The results of the API 20NE system show that the bacterium is able to synthesize nitrate, nitrite, hesperidin, D-maltose, adipic acid, malic acid, and citric acid. The results of the API ZYM system show that the strain is able to produce alkaline phosphatase and esterase-like enzymes. Amino acid aromatase, trypsin, and acid phosphatase tests show weakly positive results, while the test results for other enzymes are negative (Table 1). The detailed physiological and biochemical properties and colony characteristics of the strain are similar to *Rhodococcus* [44].

Considering that the temperature in the strain’s usual environment is constantly changing, the growth of the strains at different temperatures was examined in laboratory conditions. The results show that TAW-CT127 is able to grow in the variation range of 4–45 °C, and that the variation is greater in the range of 30–45 °C, where the optimal growth temperature of the strain is 45 °C (Figure 2a).

The initial pH of the medium is also an important factor affecting the growth of the strain, and the results show that TAW-CT127 could grow in the pH range of 5.0–8.0, with the optimum pH = 7.0 for the growth of the strain (Figure 2b).

The strain was isolated from submarine sediment samples, and seawater is rich in salts. As such, the salinity of the medium was examined among the influencing factors affecting the growth of the strain. The strains are able to grow in the salinity range of 0–8% (*w*/*v*), and the most significant difference in growth is observed at a salinity of 2.5% (Figure 2c).

Combined with the above results, this result indicates that strain TAW-CT127 does not grow and reproduce well in strongly acidic or alkaline environments, but is able to adapt to extreme environments, especially at high temperatures and salinities.

### 3.2. Molecular Biological Identification

The BLAST search results for 16S rRNA of strain TAW-CT127 (Appendix A) show genetic sequence similarity (100%) between strain TAW-CT127 and the model strain *Rhodococcus ruber* DSM 43338 (T) (accession number: X80625) (Figure 3). The strain was identified as belonging to the genus *Rhodococcus*, which is widely reported for the utilization of organic substances, including the degradation of organic pollutants such as alkanes [45,46,47,48] and aromatic hydrocarbons [49,50].

### 3.3. Study on the Degradation Characteristics of BDE-209 by Strains

#### 3.3.1. Observation of Strains Growing in Different Culture Media

In response to external stimuli, the bacteria and pollutants interact and BDE-209 is translocated intracellularly and degraded, a process that leads to a series of changes in cellular characteristics [37]. PBDEs inhibit the expression of bacteriophage proteins, thereby increasing the permeability of the cell membrane [51]. The strains were inoculated in different nutritional state media for 5 days, followed by SEM observation, which shows that TAW-CT127 is micro-rod-shaped under eutrophic conditions, and that the surface is smooth and round (Figure 4a). The bacteria in the BDE-209 degradation medium are micro-rod-shaped, with dented and wrinkled surfaces (Figure 4b). These results indicate that TAW-CT127 is damaged when grown in MMC medium with BDE-209 as the sole carbon source instead of eutrophic conditions.

#### 3.3.2. Degradation Characteristics of BDE-209 by Strain TAW-CT127

The degradation efficiency of BDE-209 is 8.80% in the MMC medium containing BDE-209 only after 5 days of incubation at 28 °C in the dark. In the experimental group inoculated with 3 g/L (wet weight) TAW-CT127 strain, the BDE-209 concentration is reduced from the initial 50 to 9.46 mg/L, a BDE-209 degradation efficiency of 81.07% (Figure 5). The metabolites of TAW-CT127 include hexabromo-, octabromo-, and nonabromo-diphenyl ethers (Figure 6). The detection of low PBDEs in the culture medium indicate that BDE-209 debromination occurs under the action of the degrading strain TAW-CT127, which is consistent with the results of a previous study [34]. Deng et al. report that a strain of *Lysinibacillus fusiformis* exhibits specific degradation of BDE-209, and octa-brominated and hepta-brominated low brominated diphenyl ethers were found in the analysis of metabolites [52]. Combined with the above information, BDE-209 can produce low brominated diphenyl ethers when metabolized by aerobic bacteria. Mai et al. analyzed the content of PBDEs in sediment samples collected from the Liangjiang River, and discovered that, although the content of BDE-209 accounted for more than 70% of the total content of all PBDEs, these lower brominated diphenyl ethers (BDE-28, -47, -66, -100, -99, -154, -153, -138, -183) were nevertheless detected in various sediment samples [26]. Considering the debromination of BDE-209 by aerobic-degrading bacteria in the laboratory, and the content distribution of PBDEs in the natural environment, aerobic-degrading microorganisms of BDE-209 play an important role in the natural metabolic transformation of BDE-209.

### 3.4. Genome Information of Strain TAW-CT127

#### 3.4.1. Genomic Properties of Strain TAW- CT127

In this study, Illumina MiSeq sequencing technology was used to complete genome scanning and sequencing of the strain, and an Illumina PE library (5000 bp) was constructed. After quality control of the sequencing data, bioinformatics analysis was performed to complete whole-genome scanning. Table 2 shows the final genomic information. TAW-CT127 was assembled into one scaffold of a 5,395,558 bp circular chromosome with a G + C content of 70.44%. There were 55 tRNAs and 1 rRNA in the TAW-CT127 genome.

#### 3.4.2. COG Database Annotation

The COG annotation results corresponding to the genes were obtained by comparison with the STRING database, and proteins were classified according to the COG annotation results. There are 5046 predicted protein sequences in the genome of TAW-CT127, of which 2846 proteins are classified by COG. The 2846 proteins are classified into four categories, and 21 types are within the COG categories A–M and O–V (Figure 7). The strain has no COG classification regarding the extracellular, nuclear, and functional proteins of the cytoskeleton and cell motility. In terms of functional classification (Appendix A), there are 250 genes related to amino acid transport and metabolism (8.78%) to regulate BDE-209 metabolism; 146 related to translation, ribosomal structure, and biogenesis (5.13%); 120 related to DNA duplication, recombination, and repair (4.22%); and 334 related to lipid transport and metabolism (11.74%). Under the stress of BDE-209, the strain produces reactive oxygen species, which may cause DNA damage. However, the main function of the genes associated with DNA replication and recombination is self-repair and reproduction of TAW-CT127. The energy production and conversion genes may be associated with the energy required for metabolic processes. In addition, 102 genes are associated with cell wall/cell membrane/envelope biogenesis (3.58%), and in Section 3.3.1, TAW-CT127 shows dents and wrinkles on the surface of the bacterium in MMC medium with BDE-209 as the only carbon source, which may be related to the expression of these genes. The percentage of unknown functional protein(s) in the COG classification is 6.96%, which is worthy of further study.

#### 3.4.3. KEGG Database Annotation

KEGG analysis shows that complete metabolic pathways are present in TAW-CT127, including pyrimidine metabolism, purine metabolism, lysine biosynthesis, the pentose phosphate pathway, and the riboflavin metabolic pathway. In addition, the glycolytic process is complete in this strain. The bacterium can produce acetone, has lactate dehydrogenase, and can act on acetone to produce lactate under anaerobic conditions, but lacks the enzyme associated with ethanol production. The tricarboxylic acid cycle metabolism pathway is also present, but only one malate dehydrogenase acts on malate to produce oxaloacetate, and this process is irreversible. This strain has a relatively complete fatty acid synthesis pathway, but lacks fatty acylcarnitine transferase in the fatty acid degradation pathway, and has a complete metabolic pathway for the production of enhanced fatty acids from alkanes. The bacterium possesses salicylate hydroxylase, which is involved in the degradation of dioxins [53], polycyclic aromatic hydrocarbons, and naphthalene [54,55].

#### 3.4.4. Analysis of BDE-209 Degradation-Related Genes

BDE-209 generates lower brominated diphenyl ethers under natural environmental conditions. Most experiments on the degradation of BDE-209 by microorganisms also show that BDE-209 undergoes a debromination reaction. Genomic analysis reveal the presence of a certain number of dehalogenases in strain TAW-CT127, and the functions of these enzymes may play vital roles in the metabolism of certain substances (Table 3). For example, in the strain TAW-CT127 genome, γ-hexachlorocyclohexane is subjected to the action of relevant functional enzymes to produce chloroacetic acid, which enters the metabolism of ethanoic acid after the production of acetic acid by the action of 2-haloacid dehalogenase (Section 3.8.1.2 in Figure 8).

In addition, genomic analysis reveals that orf00636 and orf03939 were found to be biphenyl-2,3-dioxygenase (BDO) when compared with the non-redundant protein sequence database (Table 4). Overwin et al. found BDO in *Burkholderia* sp. strain LB400, which is capable of oxidizing biphenyl and other related aromatic compounds, generates 2,3 dihydroxybiphenyl by eliminating HX (X = Br, F, or NO2) in the adjacent and interposition and further catabolizing their metabolism [56]. Since lower brominated diphenyl ethers were found in the TAW-CT127 degradation products of BDE-209, we speculate that BDO may catalyze the debromination reaction of BDE-209 in TAW-CT127.

Related functional enzymes were found in the degradation pathways of benzoic acid, nitrotoluene, and aromatic hydrocarbon compounds. BDE-209 is formed by the ether bond connecting two benzene rings; therefore it may have a similar degradation pathway to these substances. Further experiments are required to determine whether related gene fragments and functional enzymes play a role in the metabolism of TAW-CT127. In summary, this study provides a preliminary exploration of the genes found in strain TAW-CT127 that are implicated in the degradation of BDE-209. This also lays the foundation for studying the bioremediation of BDE-209 from a molecular perspective.

## 4. Conclusions

In this study, a strain of bacteria with the ability to degrade BDE-209 was isolated from sediment, and named *Rhodococcus ruber* TAW-CT127. Physiological and biochemical experiments and 16S rRNA sequencing analysis reveal that the bacterium belongs to the genus *Rhodococcus*. The growth temperature and pH range of TAW-CT127 is 4–45 °C (optimum, 45 °C) and 5.0–8.0 (optimum, pH 7.0), respectively. The optimum growth salinity is 0–2.5%. TAW-CT127 exhibits strong tolerance to high temperatures and salinity. GC–MS analysis of BDE-209 shows that under BDE-209 50 mg/L, the degradation rate is 81.07% in experimental group after 5 days, and the low-brominated products are hexabromo-, octabromo-, and nonabromo-diphenyl ethers. Genomic analysis shows that complete metabolic pathways, such as glycolysis, fatty acid synthesis, and lysine synthesis are present in strain TAW-CT127, and a certain number of dehalogenases are present that may be involved in the debromination reaction during the metabolism of BDE-209. In addition, the BDO corresponding to the sequences orf00636 and orf03939 may be involved in the debromination reaction of BDE-209. Our study isolated the strain TAW-CT127, which is an effective strain for bioremediation of BDE-209 contamination, especially in high temperature and high salinity environments, and conducted a preliminary study on microbial degradation of BDE-209 from a genomic perspective.

## Figures and Tables

**Figure 1 microorganisms-10-01441-f001:**
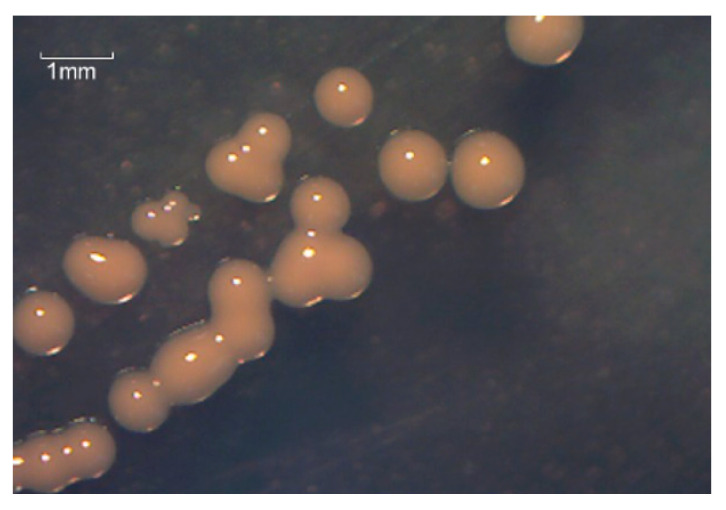
Oil micrograph of the Gram-negative strain TAW-CT127 grown marine 2216 Luria–Bertani agar at 28 °C for 72 h.

**Figure 2 microorganisms-10-01441-f002:**
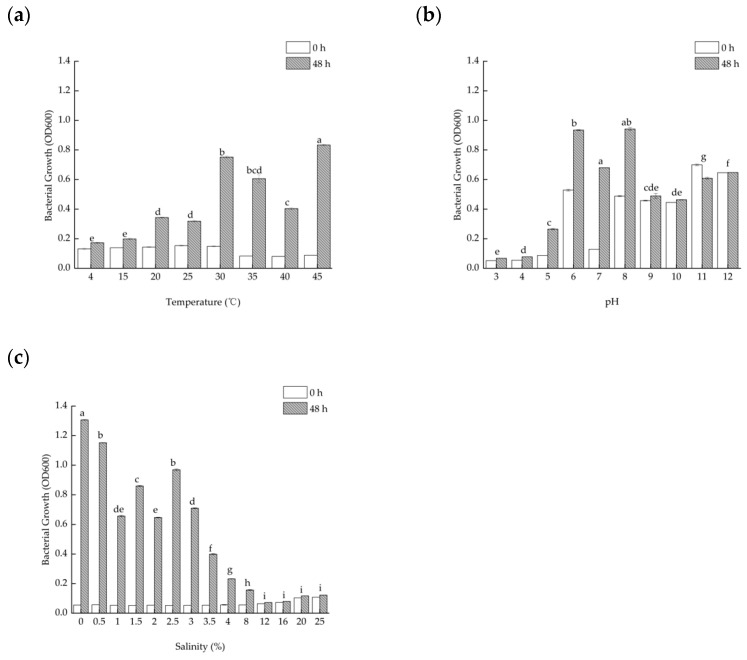
Optimal growth conditions for the strain TAW-CT127. (**a**) Growth of strain TAW-CT127 at different temperatures. (**b**) Growth of strain TAW-CT127 at different pH. (**c**) Growth of strain TAW-CT127 at different salinities. Values with completely different superscript letters in the same column are significantly different from each other, *p* < 0.05, and error bars indicate the standard error of the mean in triplicate experiments (using Tamhan correction).

**Figure 3 microorganisms-10-01441-f003:**
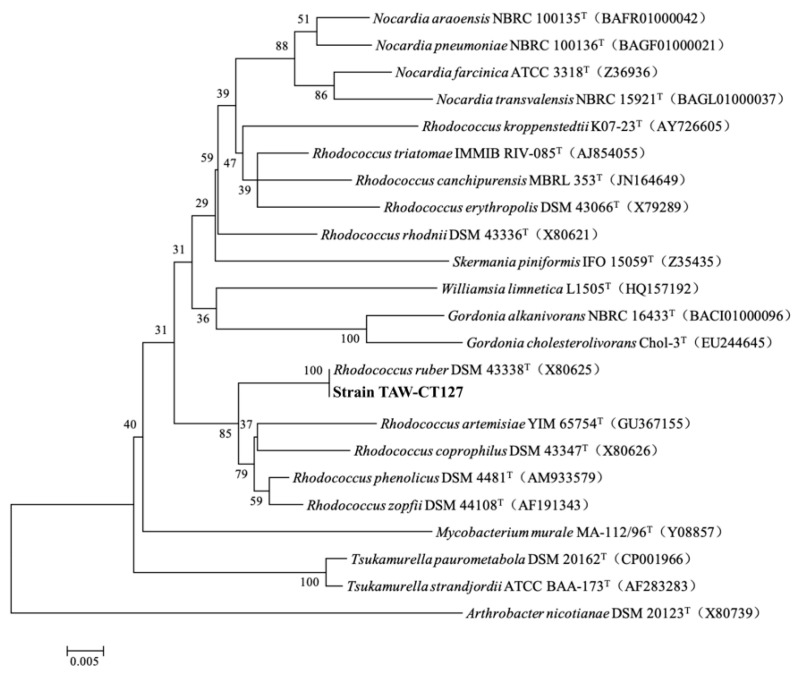
Phylogenetic trees constructed from the 16S rRNA sequences of strain TAW-CT127 and some other strains by the neighbor-joining method. The bold captions are the strain in this study.

**Figure 4 microorganisms-10-01441-f004:**
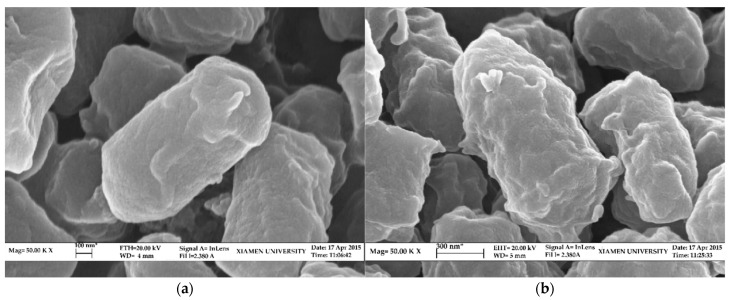
SEM images of strain TAW-CT127 in different media after 5 d culture. (**a**) SEM image of strain TAW-CT127 in the marine 2216 L–B medium. (**b**) SEM image of strain TAW-CT127 in the MMC medium.

**Figure 5 microorganisms-10-01441-f005:**
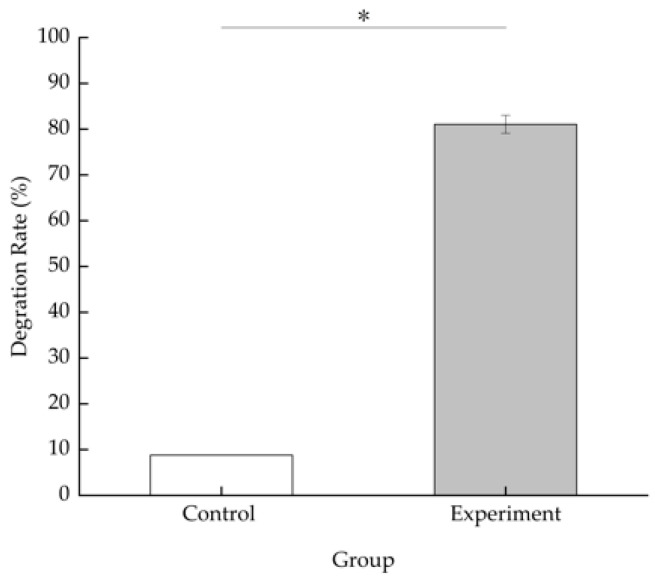
Degradation efficiency of BDE-209 by strain TAW-CT127 at 3 g/L (wet weight), 28 °C, pH 7.4 ± 0.2, 160 rpm for 5 d, *: *p* < 0.05.

**Figure 6 microorganisms-10-01441-f006:**
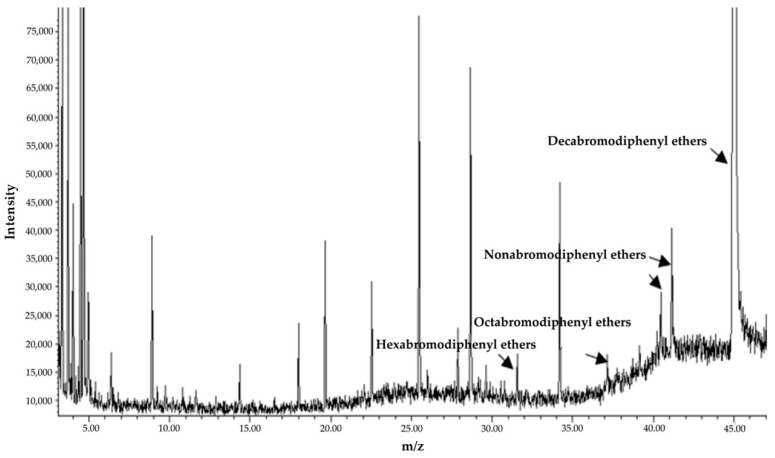
Total ion chromatogram of BDE-209 degrading culture (TAW-CT127) extract.

**Figure 7 microorganisms-10-01441-f007:**
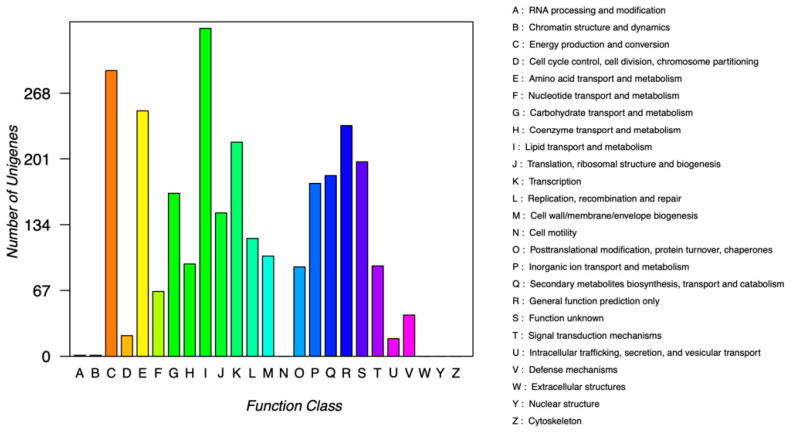
COG functional classification of strain TAW-CT127.

**Figure 8 microorganisms-10-01441-f008:**
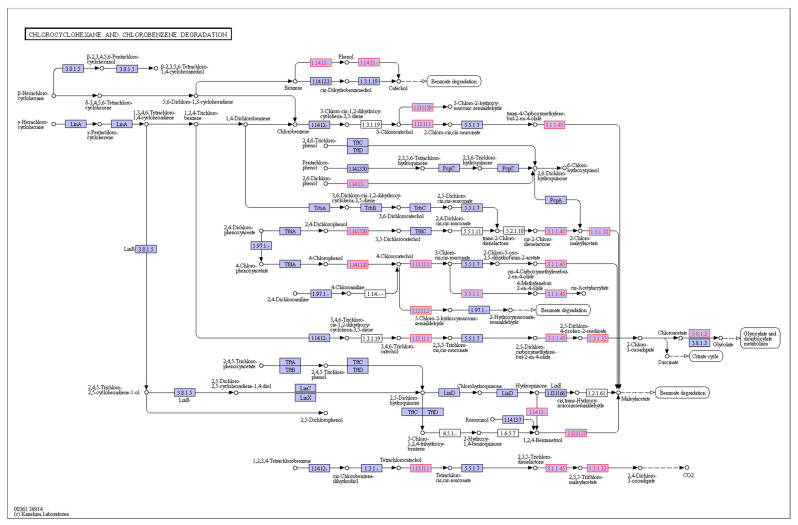
Chlorocyclohexane and chlorobenzene degradation in TAW-CT127.

**Table 1 microorganisms-10-01441-t001:** Phenotypic characteristics of strain TAW-CT127.

Characteristics	Results	Characteristics	Results
Physiology and Biochemistry		Arginine dihydrolase	−
Gram stain	+	Qiyeling hydrolysis	+
Starch hydrolysis	−	D-maltose	+
V–P	−	Decanoic acid	−
Catalase	+	Adipic acid	+
Contact oxidase	−	Malic acid	+
API 20E		Citric acid	+
Arginine	−	Phenylacetic acid	−
Dihydrolase	−	API ZYM	
Lysine decarboxylase	−	Alkaline phosphatase	+
Ornithine decarboxylase	+	Esterase (C4)	−
The generation of H_2_S	−	Lipoesterase (C8)	−
Urea enzyme	−	Lipase (C14)	W
Tryptophan deaminase	−	Leucine arylamidase	+
Indole production	−	Valine arylase	W
Production of acetylmethyl methanol	+	Cystine arylamidase	−
Gelatinase	−	Trypsin	W
Glucose fermentation	−	Acid phosphatase	W
Sucrose fermentation	−	Naphthol-AS-BI-phosphate hydrolase	−
API 20NE	−	α-galactosidase	−
Nitrate reduction (NO^3-^)	+	β-galactosidase	−
Nitrate reduction (NO^2-^)	+	α-glucosidase	+
Indole production	−	β-glucosidase	−
Glucose acidification	−	N-acetyl-β-glucosaminidase	−

**Note.** Characteristics are scored as: W, weak; +, positive; −, negative.

**Table 2 microorganisms-10-01441-t002:** Genome information of strain TAW-CT127.

Categories	Results	Categories	Results
Bases in all scaffolds	5,395,558 bp	Contig N50	15,825 bp
G + C content	70.44%	Gene num	5046
No. of large scaffolds (>1000 bp)	440	Gene total length	4,603,239 bp
Scaffold N50	22,120 bp	tRNA	55
No. of large contigs (>1000 bp)	562	rRNA	1

**Table 3 microorganisms-10-01441-t003:** Dehalogenases in the genome of strain TAW-CT127.

Stain	Orf Number	Dehalogenase	Microorganism	Similarity
TAW-CT127	orf02266	Aloalkane dehalogenase	*Rhodococcus* sp. P14	100%
TAW-CT127	orf02896	Haloacid dehalogenase	*Rhodococcus*	100%
TAW-CT127	orf02614	Haloacid dehalogenase	*Rhodococcus*	100%
TAW-CT127	orf02255	Haloacid dehalogenase	*Rhodococcus ruber*	100%
TAW-CT127	orf03435	Haloacid dehalogenase	*Rhodococcus* sp. P14	100%
TAW-CT127	orf04248	Haloacid dehalogenase	*Rhodococcus* sp. P14	100%
TAW-CT127	orf01345	Haloacid dehalogenase	*Rhodococcus*	100%
TAW-CT127	orf04277	Haloacid dehalogenase	*Rhodococcus ruber*	100%
TAW-CT127	orf02806	Haloacid dehalogenase	*Rhodococcus ruber*	98%
TAW-CT127	orf00660	Haloacid dehalogenase	*Rhodococcus*	100%

**Table 4 microorganisms-10-01441-t004:** Biphenyl-2,3-dioxygenases in the genome of strain TAW-CT127.

Stain	Orf Number	Description	Microorganism	Similarity
TAW-CT127	orf00636	Biphenyl-2,3-dioxygenase	*Rhodococcus ruber*	100%
TAW-CT127	orf03939	Biphenyl-2,3-dioxygenase	*Rhodococcus*	100%

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
