# Peer review of "Aerobic Degradation Characteristics of Decabromodiphenyl ether through Rhodococcus ruber TAW-CT127 and Its Preliminary Genome Analysis"

_microorganisms, 2022, doi:10.3390/microorganisms10071441_

Round 1

Reviewer 1 Report

The manuscript is devoted to removing persistent organic pollutants from natural and man-made environments. Decabromodiphenyl ether (BDE-209)––a polybrominated diphenyl ethers (PBDEs) homolog––poses a threat to ecosystems, living organisms and accumulates within food chain in significant quantities. This compound is resistant to physicochemical degradation, which contributes to its long-term presence in natural matrices. Even though a significant amount of published data is devoted to the problem of its neutralization, basic issues related to the search for optimal environmentally friendly and economically viable approaches to removal of BDE-209 from the environment still remain open. In this regard, one of the promising directions in solving this problem is the study of the degradative potential of natural aerobic bacteria. Bacteria play the role of the primary response system to the xenobiotic load and initiate adaptive reactions at the earliest development stages. It is not surprising that among the promising biodegraders of BDE-209, the authors pay special attention to representatives of genus Rhodococcus (Atinomycetia class), which have the greatest variety of degradable pollutants.

The manuscript is structured in accordance with the requirements of the journal, the list of cited literary sources is convincing.

General comments:

1. The manuscript title is confusing. I assume that it is supposed to be “Aerobic Degradation Characteristics…”

2. Rhodococci are Gram-positive, not Gram-negative bacteria. When staining according to Gram, it is necessary to consider the age of the culture.

Author Response

Point 1: The manuscript is devoted to removing persistent organic pollutants from natural and man-made environments.

Response 1: Thank you for your affirmation for our manuscript.

Point 2: The manuscript title is confusing. I assume that it is supposed to be “Aerobic Degradation Characteristics…”

Response 2: Thank you so much for your comprehensive evaluation. The strain in this study was cultured under aerobic conditions. And Rhodococci are non-sporulating, aerobic bacteria classified into mycolate-containing nocardioform actinomycetes [1]. According to your questioning, we have revised the title of the manuscript, as " Aerobic Degradation Characteristics of Decabromodiphenyl ether through Rhodococcus ruber TAW-CT127 and Its Preliminary Genome Analysis ". Also, we have again upgraded the language and revised the full text carefully. Please check the re-submitted manuscript for details.

Point 3: Rhodococci are Gram-positive, not Gram-negative bacteria. When staining according to Gram, it is necessary to consider the age of the culture.The manuscript title is confusing. I assume that it is supposed to be “Aerobic Degradation Characteristics…”

Response 3: Thank you very much and we took your suggestion and re-conducted the experiment of Gram staining. According to the results, Rhodococcus ruber TAW-CT127 was Gram-positive. The inconsistent results confused us a lot. After the comprehensive literature research, we propose a hypothesis that might explain the discrepancy. Bacteria with a thick, highly cross-linked layer of peptidoglycan trap the primary stain-mordant complex and stain purple, and are designated Gram-positive. Those bacteria that have a thin layer of peptidoglycan with a lower percentage of cross-linkage, followed by a thin second layer called the outer membrane, do not retain the primary stain-mordant complex upon alcohol treatment. These bacteria with a thin cell wall are counterstained with safranin and are labeled as the Gram-negative bacteria. Since the Gram stain relies on an intact cell wall, a log phase (actively growing) culture is required. Older cultures will give Gram variable results due to degradation of the cell wall [2].

In summary of the above-mentioned mechanisms, we believed that might be a possible explanation to cover this issue, and line 704 and table 1 have been modified based on your valuable suggestion. Thanks again for your advice.

References of the responses to reviewer 1

  1. Martínková, L.; Uhnáková, B.; Pátek, M.; Nesvera, J.; Kren, V. Biodegradation Potential of the Genus Rhodococcus. Environ Int 2009, 35, 162–177, doi:10.1016/j.envint.2008.07.018.
  2. Moyes, R.B.; Reynolds, J.; Breakwell, D.P. Differential Staining of Bacteria: Gram Stain. Curr Protoc Microbiol 2009, Appendix 3, Appendix 3C, doi:10.1002/9780471729259.mca03cs15.

Reviewer 2 Report

The reviewed paper is preliminary research. Moreover, in the manuscript are so many methodological errors and erroneous assumptions that the results are questionable. To name just a few:

-The title of the manuscript suggests that the authors investigated the degradation of BDE-209, but there is no evidence for this. In Figure 5, the authors presented the degradation efficiency of BDE-209, which shows that abiotic degradation (control) is more efficient (about 95% efficiency) than in the biotic system (79.24%). The entire experiment is poorly planned. Hence, the results do not support the conclusions about the degradation of BDE-209 by the tested strain.

- Concentrations (50 and 100 mg/L) of decabromodiphenyl ether (BDE-209) were used that were well above its solubility (about 10 µg/L), so the substrate was not available for bacteria; BDE is a volatile compound, which the authors did not take into account in the research; moreover, in such heterogeneous systems it is not possible to test bacterial growth via optical density;

-  The descriptions of some experiences are inaccurate, and all of them cannot be replicated. For example, it is unknown what medium was used to determine the optimal growth conditions of the tested strain.

-  Determination of the optimal growth conditions for the tested strain. It is clearly seen in Figure 2 that at time 0, the authors did not use the same optical density for the culture. Hence, comparing the results between the tested systems is not allowed.

-  It is unknown on what basis the authors claim that they investigated the molecular basis of BDE-209 degradation. The authors did not specify the activity of any enzymes. Moreover, as intermediates, they list only a few intermediates with a reduced number of bromo substituents. It should be noted that the authors did not include any spectra of the identified intermediates in the manuscript, which is a standard in good papers. Moreover, the authors do not present evidence of aromatic rings' cleavage. They only analyze the genes of the strain under study. However, the presence of genes is not evidence of their activity. For this, it is necessary to determine at least the transcriptional activity of such genes.

- Figure 7 - It is unknown on what basis the authors propose the path of chlorocyclohexane and chlorobenzene decomposition by the studied strain. The authors did not demonstrate the presence of any of the intermediates of this pathway nor the enzyme's activity. Genes in the genome do not prove that the metabolic pathway is active.

- The presented conclusions were not supported by the results.

-   Moreover, the manuscript is written in very bad English, including the title of the manuscript, with numerous typos and linguistic and grammatical errors.

-  In conclusion, the paper is full of methodical errors, contains questionable results and, in my opinion, should not be published.

Author Response

Point 1: The title of the manuscript suggests that the authors investigated the degradation of BDE-209, but there is no evidence for this. In Figure 5, the authors presented the degradation efficiency of BDE-209, which shows that abiotic degradation (control) is more efficient (about 95% efficiency) than in the biotic system (79.24%). The entire experiment is poorly planned. Hence, the results do not support the conclusions about the degradation of BDE-209 by the tested strain.

Response 1: Thank you for your valuable comments on our manuscript. â‘  This study was conducted based on the experimental method of Tang et al. [1]. The Rhodococcus ruber TAW-CT127 strain was obtained by screening soil samples by adding them to MMC medium with BDE-209 as the sole carbon source. Therefore, this strain is able to use BDE-209 as a carbon source and consume it. â‘¡ In the original manuscript, we directly expressed the degradation rate results of the experimental group as 79.24% of the measured value. As you remind, we believe that the natural degradation rate (8.08%) of BDE-209 in the control group should indeed be taken into account. Therefore, we express the degradation rate of the experimental group as 81.07%. Therefore, the Rhodococcus ruber TAW-CT127 strain has the ability to degrade BDE-209. Lines 501-506, 792-795 and figure 5 have been updated in response to your suggestions. 

Point 2: Concentrations (50 and 100 mg/L) of decabromodiphenyl ether (BDE-209) were used that were well above its solubility (about 10 µg/L), so the substrate was not available for bacteria; BDE is a volatile compound, which the authors did not take into account in the research; moreover, in such heterogeneous systems it is not possible to test bacterial growth via optical density.

Response 2: Please accept our heartfelt gratitude for your question, which led us to understand that the description in the initial manuscript was confusing.

â‘  Firstly, strain TAW-CT127 was obtained by screening in MMC medium supplemented with 100 mg/L of BDE-209 as the sole carbon source, indicating that the strain is able to utilize BDE-209 as a carbon source during growth and metabolism. Secondly, Tang et al. screened the strain using MSM medium containing up to 100 mg/L BDE-209 [1]. Lu et al. used MMC medium containing 20 mg/L BDE-209 for culturing the strain when they studied the biodegradation characteristics of Bacillus cereus JP12 towards BDE-209 [2]. Paliya et al. used an initial BDE-209 Bacillus tequilensis BDE-S1 bacteria using MSM medium with an initial BDE-209 concentration of 50 mg/L and determined the degradation ability of this strain [3]. Similarly Tang et al. studied the degradation characteristics of BDE-209 by Brevibacillus brevis with an initial concentration of BDE-209 of 0.5 mg/L [4]. The concentrations of BDE-209 in the above studies were well above its solubility (about 10 µg/L), indicating that the bacteria are can utilize BDE-209 in the medium.

â‘¡ We are sorry that we may have caused you such confusion due to our incomplete description of the experiment details. Our experimental procedure was to weigh 10 mg of BDE-209 into a 100 ml volumetric flask and add chromatography grade dichloromethane. Dissolution was assisted by sonication for 30 min, adding ice continuously during this process to prevent loss of BDE-209 from increased temperature. Then make up to a BDE-209 stock solution at a concentration of 100 mg/L. Add an appropriate amount of stock solution to a 50 ml sterile centrifuge tube and wait until the dichloromethane is completely evaporated. The control group was only supplemented with MMC medium, and the experimental group was supplemented with MMC medium and TAW-CT127 strain resuspension in section 2.5 so that the initial concentration of BDE-209 was 50 mg/L. The above centrifuge tubes were incubated at 28 °C, 160 rpm for 5 days under protected from light. All the above procedures were operated under light-proof conditions. After reviewing the literature on microbial remediation of BDE-209, our experimental procedure was not much different from other literature [3–5], and we kept low temperature and light-proof conditions during the experiment, which effectively prevented the natural degradation of BDE-209. In this study both the precision and the recovery of the method were investigated. The results showed that the relative guarantee deviation of precision was 2.1%. The recoveries of the method ranged from 90.0% to 92.0%.

â‘¢ We sincerely apologize for not presenting enough details of the experiments in the manuscript again. Experiments regarding the optimal growth conditions of TAW-CT127 were performed by inoculating the strain resuspension in marine 2216 LB medium at 3 g/L (wet weight), and measure the optical density of the bacterial solution at OD600 using a full-wavelength enzyme standard. Similarly, Tang et al. used marine 2216 LB medium to culture the strain and determined optical density at a wavelength of 600 nm to determine the growth conditions of the bacteria [5]. Lines 282-284 have been updated in response to your suggestions. Therefore, inoculation of TAW-CT127 in marine 2216 LB medium and optical density testing of bacterial growth was not performed in a heterogeneous system. We apologize again for the insufficient description of some of the details, and hope the above answer will satisfy you.

Point 3: The descriptions of some experiences are inaccurate, and all of them cannot be replicated. For example, it is unknown what medium was used to determine the optimal growth conditions of the tested strain.

Response 3: Thank you for your kind suggestion and we are sorry for our insufficient description in the original manuscript. We are happy to provide the necessary experimental details to ensure reproducibility of the experiments and inoculated the resuspension of strain TAW-CT127 with marine 2216 LB medium at an inoculum level of 3 g/L (wet weight) and measure the optical density of the bacterial solution at OD600 using a full-wavelength enzyme standard at 0 and 48 hours respectively. Lines 294-296, 308-316, 487-488, 498-501 and 515-522 have been updated in response to your suggestions. Based on your comments, we have explained this experimental procedure in more detail to avoid confusion among readers.

Point 4: Determination of the optimal growth conditions for the tested strain. It is clearly seen in Figure 2 that at time 0, the authors did not use the same optical density for the culture. Hence, comparing the results between the tested systems is not allowed.

Response 4: We heartily thank the reviewer for raising concerns about the optimal growth conditions for the tested strain. We inoculated the resuspension of strain TAW-CT127 with marine 2216 LB medium at an inoculum level of 3 g/L (wet weight) [6] and measure the optical density of the bacterial solution at OD600 using a full-wavelength enzyme standard at 0 and 48 hours respectively. During the experiment, we used an inoculation loop to pick a single colony of TAW-CT127 from the marine 2216 agar plate, and then inoculated into the marine 2216 medium for overnight culture. Next make a strain resuspension as in Section 2.5. Due to the logarithmic growth of bacteria and some inevitable systematic errors in the process of using the inoculation loop, the optical density of the bacterial solution at 0 h in Figure 2 is different. We have redrawn figure 2a and changed line 748.

Point 5: It is unknown on what basis the authors claim that they investigated the molecular basis of BDE-209 degradation. The authors did not specify the activity of any enzymes. Moreover, as intermediates, they list only a few intermediates with a reduced number of bromo substituents. It should be noted that the authors did not include any spectra of the identified intermediates in the manuscript, which is a standard in good papers. Moreover, the authors do not present evidence of aromatic rings' cleavage. They only analyze the genes of the strain under study. However, the presence of genes is not evidence of their activity. For this, it is necessary to determine at least the transcriptional activity of such genes.

Response 5: We are all very appreciated for this professional suggestion. First, based on our experimental results which found that the metabolites of BDE-209 by strain TAW-CT127 included hexabromo-, octabromo-, and nonabromo-diphenyl ethers, we speculate that BDE-209 underwent dehalogenation during the metabolism of the strain, but the metabolic mode is not yet clear. Secondly, our genomic analysis of strain TAW-CT127 revealed the presence of a certain number of dehalogenases, among which 2-haloacid dehalogenases (3.8.1.2) are present in the degradation pathway of chlorocyclohexane and chlorobenzene. Namely, γ-hexachlorocyclohexane produced chloroacetic acid by the action of related functional enzymes, and subsequently entered into the metabolism of ethanoic acid after the production of acetic acid by the action of 2-haloacid dehalogenases. We only take the 2-haloacid dehalogenases involved in the chloroacetic acid dehalogenation reaction as an example to initially explore the enzymes related to the degradation process of BDE-209, and the study of related enzyme activities and metabolic pathways is the focus of subsequent work. Lines 935-938 and 950-954 have been updated in response to your suggestions.

Point 6: The presented conclusions were not supported by the results.

Response 6: Thank you really much for your suggestions. We have carefully reworked the conclusions of the manuscript in accordance with your suggestions and have made changes in section 4 “Conclusions”.

Point 7: Moreover, the manuscript is written in very bad English, including the title of the manuscript, with numerous typos and linguistic and grammatical errors.

Response 7: Thanks a lot for this kind reminding from reviewer. We revised the title of the manuscript, changed it to "Aerobic Degradation Characteristics of Decabromodiphenyl ether through Rhodococcus ruber TAW-CT127 and Its Preliminary Genome Analysis". In addition, we had the original manuscript touched up by a professional firm before submission. Based on your suggestions, we have again upgraded the language and revised the full text. We hope the revised manuscript would satisfy you.

Point 8: In conclusion, the paper is full of methodical errors, contains questionable results and, in my opinion, should not be published.

Response 8: Our group mates are sincerely appreciated for your careful and detailed review comments. Your professional suggestions proposed us to realize there were still space of this study and deserved further in-depth exploration. We will continue to improve the study in the follow-up.

After re-checking the full text of our initial manuscript, we have tried our best to revised the misleading and non-standard descriptions, and answered your comments carefully one by one. Maybe our work did not meet your requirements, we still worked hard to make it better, because we all believed our work not only fell within the scope of Microorganisms, but also had potential applications for environmental protection.

We sincerely hope that you can re-evaluate this article. Thank you very much for your valuable time in reviewing the manuscript.

References of the responses to reviewer 2

  1. Tang, S.; Yin, H.; Chen, S.; Peng, H.; Chang, J.; Liu, Z.; Dang, Z. Aerobic Degradation of BDE-209 by Enterococcus Casseliflavus: Isolation, Identification and Cell Changes during Degradation Process. J Hazard Mater 2016, 308, 335–342, doi:10.1016/j.jhazmat.2016.01.062.
  2. Lu, M.; Zhang, Z.-Z.; Wu, X.-J.; Xu, Y.-X.; Su, X.-L.; Zhang, M.; Wang, J.-X. Biodegradation of Decabromodiphenyl Ether (BDE-209) by a Metal Resistant Strain, Bacillus Cereus JP12. Bioresour Technol 2013, 149, 8–15, doi:10.1016/j.biortech.2013.09.040.
  3. Paliya, S.; Mandpe, A.; Kumar, M.S.; Kumar, S. Aerobic Degradation of Decabrominated Diphenyl Ether through a Novel Bacterium Isolated from Municipal Waste Dumping Site: Identification, Degradation and Metabolic Pathway. Bioresource Technology 2021, 333, 125208, doi:https://doi.org/10.1016/j.biortech.2021.125208.
  4. Tang, S.; Bai, J.; Yin, H.; Ye, J.; Peng, H.; Liu, Z.; Dang, Z. Tea Saponin Enhanced Biodegradation of Decabromodiphenyl Ether by Brevibacillus Brevis. Chemosphere 2014, 114, 255–261, doi:10.1016/j.chemosphere.2014.05.009.
  5. Tang, C.; Fan, C.; Guo, D.B.; Ma, X.J.; Cai, Q.T.; Chen, X.X.; Zhang, M.; Li, J.Y.; An, Q.Y.; Zhao, R. Identification of Boseongicola Sediminum Sp. Nov., a Novel Decabromodiphenyl Ether (BDE-209)-Tolerant Strain Isolated from Coastal Sediment in Xiamen, China. Biomed Environ Sci 2021, 34, 656–661, doi:10.3967/bes2021.092.
  6. Zhao, R.; Wang, B.; Cai, Q.T.; Li, X.X.; Liu, M.; Hu, D.; Guo, D.B.; Wang, J.; Fan, C. Bioremediation of Hexavalent Chromium Pollution by Sporosarcina Saromensis M52 Isolated from Offshore Sediments in Xiamen, China. Biomed Environ Sci 2016, 29, 127–136, doi:10.3967/bes2015.014.

Round 2

Reviewer 2 Report

The authors corrected and supplemented the manuscript with the necessary information. It can now be published.